# Long-Term Person Re-Identification Based on Appearance and Gait Feature Fusion under Covariate Changes

**Xiaoyan Lu** [1] , **Xinde Li** [1,2,3,*] **, Weijie Sheng** [3] **and Shuzhi Sam Ge** [4]

1 School of Cyber Science and Engineering, Southeast University, Nanjing 210002, China; 230189755@seu.edu.cn
2 Guangdong Intelligent Robotics Institute, Dongguan 523002, China
3 Key Laboratory Measurement and Control of CSE Ministry of Education, School of Automation,
  Southeast University, Nanjing 210002, China; wjsheng@seu.edu.cn
4 Social Robotics Laboratory, Department of Electrical and Computer Engineering,
  Interactive Digital Media Institute, National University of Singapore, Singapore 119077, Singapore;
  elegesz@nus.edu.sg
* Correspondence: xindeli@seu.edu.cn

**Abstract:** Person re-identification(Re-ID) technology has been a research hotspot in intelligent video surveillance, which accurately retrieves specific pedestrians from massive video data. Most research focuses on the short-term scenarios of person Re-ID to deal with general problems, such as occlusion, illumination change, and view variance. The appearance change or similar appearance problem in the long-term scenarios has has not been the focus of past research. This paper proposes a novel Re-ID framework consisting of a two-branch model to fuse the appearance and gait feature to overcome covariate changes. Firstly, we extract the appearance features from a video sequence by ResNet50 and leverage average pooling to aggregate the features. Secondly, we design an improved gait representation to obtain a person's motion information and exclude the effects of external covariates. Specifically, we accumulate the difference between silhouettes to form an active energy image (AEI) and then mask the mid-body part in the image with the Improved-Sobel-Masking operator to extract the final gait representation called ISMAEI. Thirdly, we combine appearance features with gait features to generate discriminative and robust fused features. Finally, the Euclidean norm is adopted to calculate the distance between probe and gallery samples for person Re-ID. The proposed method is evaluated on the CASIA Gait Database B and TUM-GAID datasets. Compared with state-of-the-art methods, experimental results demonstrate that it can perform better in both Rank-1 and mAP.

**Keywords:** person Re-ID; covariate changes; ISMAEI; appearance feature; feature-level fusion

## 1. Introduction

The goal of person re-identification (Re-ID) is to find specific targets in surveillance videos, assisting in criminal investigation, cross-camera target tracking, and behavior analysis [1–4], which can significantly improve the efficiency of social security. In practical application, people may be caught by different cameras and on various lighting conditions. Additionally, the same person's external covariates may change during long-term monitoring, such as clothing, bag, and so on. These situations will bring challenges to the person Re-ID tasks like in Figure 1.

For the aim of solving these challenges, many researchers made a lot of contributions to the task. These methods generally include person Re-ID based on appearance alone [5–7], gait alone [8,9], or a fusion of the two features [10,11]. Appearance features based on deep learning represent more informative information than the hand-crafted feature. The methods based on deep learning Appearance features could better deal with illumination change and view variant challenges. Only employing appearance features will lead to a low matching accuracy rate of person Re-ID when different people wear similar clothes or a person carries different bags. The gait representation can overcome the problem of appearance changes

because human gait representations record posture and behavior characteristics while walking and are not affected by appearance changes. Although the gait-based algorithms have the advantage of appearance invariance, they still face challenges, such as view variants, occlusion, and incomplete gait cycle. Therefore, some researchers [10,12,13] considered the two kinds of features to complement each other through the fusion method. These methods primarily combine hand-crafted appearance features with gait energy image (GEI) features. However, low-level and mid-level features are less robust to image translation, and GEI features based on image segmentation still contain noise edge, e.g., the outline of Backpacks, for person Re-ID under covariate changes. Meanwhile, the search accuracy of these methods needs to be improved.

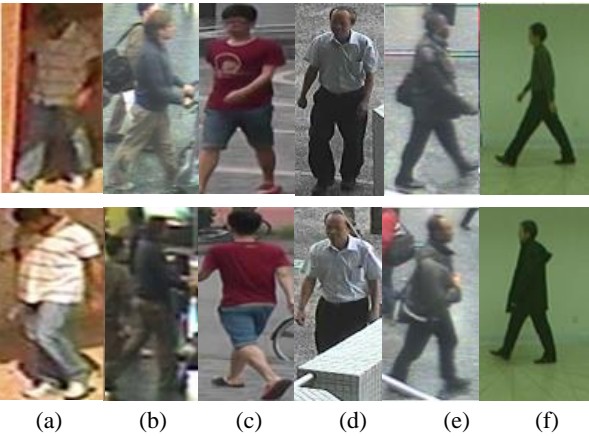

(a)　　(b)　　(c)　　(d)　　(e)　　(f)

**Figure 1.** Person Re-ID challenges: (**a**) low resolution (3DPes dataset [14]), (**b**) illumination (iLIDS-VID dataset [15]), (**c**) viewpoint variant (Market1501 dataset [16]), (**d**) occlusions(CUHK03 dataset [17]), (**e**) similar apperance (iLIDS-VID dataset [15]), (**f**) appearance changed (CASIA Gait Database B [18]).

We put forward a two-branch Re-ID model to optimize the appearance and gait feature and design fusion mode to integrate the appearance and gait features better. The detailed proposed framework can be seen in Figure 2. We first use a ResNet-50 backbone to extract the appearance features and aggregate the features by average pooling. The appearance features contain rich information, including pedestrians' shape, posture, and accessories. Secondly, We employ an Improved-Sobel-Masking operator to mask the active energy image(AEI) to filter out the noises. The improved gait features are an accumulative energy image of the human gait cycle and normalize the spatial-temporal information to characterize a human's unique properties. Then, we design a fusion method that cascades the two features with adjustment factor $\theta$. Experiments are conducted on the CASIA Gait Database B [18] and TUM-GAID dataset [19] to evaluate the proposed method. It can be seen from the experiment results that our method can exceed the state-of-the-art methods under covariate changes. In summary, the contributions of this paper can be concluded as follows:

- We propose a new two-branch Re-ID model to combine appearance information with gait information. The complementary combination ensures the uniqueness of biometrics and maintains the robustness to appearance changes.
- We put forward a novel gait representation, namely Improved-Sobel-Masking active energy image (ISMAEI), instead of common GEI, which can retain the uniqueness of human motion information and overcome covariate changes.
- We design a feature-level fusion method with an adjustment factor to better integrate appearance features with gait features.

The rest of the paper is organized as follows. Section 2 introduces the overview of related work about person Re-ID. Section 3 presents overall the proposed framework and describes the designed method of the different parts of the person Re-ID system in detail. In Section 4, the results of ablation experiments and contrast experiments are given along

with discussions. Finally, we conclude the whole work and discuss the future work in Section 5.

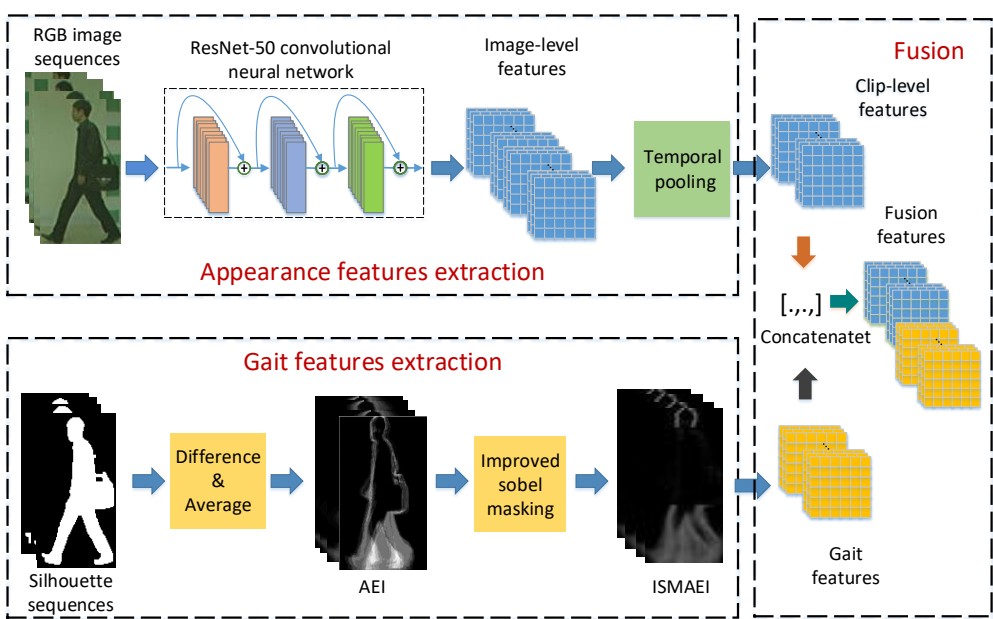

**Figure 2.** The framework of the proposed method includes three parts: appearance feature branch, gait feature branch, and feature fusion. AEI denotes active energy image, and ISMAEI represents Improved-Sobel-Masking active energy image.

## 2. Related Work

Currently, most works conduct the survey from the representation perspective. Therefore, we summarize work about Re-ID methods from three aspects: appearance-based, gait-based, and the combination of the two features.

### 2.1. Appearance-Based Person Re-ID

The concept of person Re-ID was firstly established by Gheissari et al. [20] in 2006. Since then, a mass of studies have been carried out on person Re-ID. Most appearance-based works are divided into two categories: person Re-ID based on hand-crafted features and person Re-ID based on deep-learning features.

**Methods based on hand-crafted features.** The color features and texture features are commonly used in a hand-crafted feature system. Fendri et al. [21] proposed a versatile appearance-based Re-ID method that contains three aspects of features modeling: (1) the overall chromatic content, (2) the spatial arrangement of colors into stable regions, and (3) the presence of recurrent local motifs with high entropy. These extracted features are robust against the low resolution, occlusions, and pose variants, but the process of features modeling is complex and time-consumed. D Gray et al. [22] put forward the ensemble containing color and texture channels and divided the human image into horizontal stripes according to prior knowledge. The partition operations reduce the influence of the irrelevant part, and lower computation complexity. Many subsequent works [23–25] also adopted the same set of features as [22]. Compared to the low-level color and texture features, the attributed-based features viewed as mid-level features are more robust to image translation than low-level features. Layne et al. [26] defined the space of fifteen binary attributes and trained the model to learn an attribute-centric, parts-based feature representation to solve the occlusion problem. Chi Su et al. [27] proposed Multi-Task Learning with Low-Rank Attribute Embedding (MTL-LORAE), which integrates low-level features with mid-level feature attributes to address the problem of person Re-ID on multi-cameras. Although the matching accuracy rate is high, it needs to retrain the model with each additional new sample added, which is time-consuming. A large-scale attributed-

based person Re-ID dataset was collected by Li et al. [28] and is the benchmark dataset for attribute-based person Re-ID methods.

**Methods based on deep learning features.** The development of deep learning improves the accuracy of the person Re-ID task. Researchers designed the person Re-ID methods based on deep learning from the perspective of deep learning features and loss functions. Deep-learning features include global features and local features, which are extracted directly from the whole image by CNN. The loss functions could guide the model to mine robust global features. Geng et al. [29] proposed the classification sub-network predicting the persons' ID and verification subnet, which verified whether the two images belong to the same person. An original feature extraction model called Feature Fusion Net (FFN) for person Re-ID was put forward by Wu et al. [30]. It utilized the hand-crafted features to compensate CNN features for forming a more discriminative and compact deep feature. In order to minimize the gap of the domain and learn generic feature representations, Xiao et al. [31] designed domain guided drop algorithm to optimize the learning procedure. In [32,33], the authors designed a variant of the triplet loss to perform an end-to-end deep metric learning framework. It outperformed most other algorithms which only used surrogate losses (classification, verification).

Global feature-based person Re-ID methods suffer from significant challenges such as background clutter, illumination, and pose variation while local features methods rely on more reliable prior information and avoid the limitations of global features. The core idea of the methods is to design a means of extraction and alignment of local features. Kalayeh et al. [34] proposed the Human Semantic Parsing method to extract local features from the human body. Zhao et al. [35] designed Spindle Net, which is based on 14 key points of the body to extract the local feature. In addition to the above-mentioned methods, images horizontal segmentation is a common method [36–38]. Varior et al. [36] integrated long-short-term memory network (LSTM) into a Siamese model. The architecture can process image patches sequentially and leverage the contextual information to generate more discriminative local features. Zhang et al. [37] put forward a novel aligned Re-ID method that joint learning of global and local features. Local features learning is to calculate the shortest path between two sets of local representations to align image patches, which benefits global features learning. Experimental results indicated that the performance of aligned Re-ID suppressed the human-level performance. Zheng et al. [39] jointed discriminative and generative learning in a unified network called DG-Net to deal with significant intra-class variations across different cameras.

To sum up, the above person Re-ID methods are basically based on appearance feature extraction. However, when persons with similar appearance or the same person under different covariates, such as clothing, bags, and so on, the performance of appearance-based person Re-ID degrades seriously.

### 2.2. Gait-Based Person Re-ID

Although appearance-based Re-ID methods are universal and applied widely, these methods are based on the "Appearance constancy hypothesis" and hardly deal with appearance change. In order to solve the limitation, gait-based Re-ID methods are presented. Human gait features, which are soft biometric, are the classical human movement feature in surveillance spaces. There are some advantages to person Re-ID, namely measured at a far distance, hard to fake, and unobtrusive and disadvantages, such as varying with illness, aging, and emotional states. Considering gait feature extraction and analysis, we categorize the gait-based person Re-ID methods into model-free and model-based methods [40]. Model-free approaches always directly extract information from gait images sequences (such as silhouette shape, optical flow) as gait features. Model-based approaches utilize the intermediate 2D or 3D geometric models to define the human dynamic features.

**Model-free methods.** Compared with model-based methods, model-free approaches are easily accessed and without building structural models. Since the height feature is rotation invariant and robust to changed appearance, John et al. [41] proposed a gait feature

that corresponds to the frequency response of the height temporal information and used maximum likelihood classification to identify the test person. The extraction of human silhouettes is generally tricky and the optical flow-based methods are adopted to gait features representation. Cartro et al. [42] designed a new optic flow-based descriptor, namely Pyramidal Fisher Motion, by Divergence-Curl-Shear. xm presented a Histogram of Flow Energy Image (HEFI) as a Spatio-temporal gait representation for long-term surveillance in which the appearance cues are highly volatile. Wang et al. [44] comprised optical flow energy profiles, color, and HOG3D as space-time feature representation, which automatically decompose rough video sets of pedestrians into multi segments. Comparative evaluations indicated that it can achieve the performance of state-of-the-art methods on images sequence matching and gait recognition. There are also several methods [45–47] based on the improved GEI features for overcoming covariate changes. Wei et al. [45] proposed a novel Swiss-system based on cascade ranking model (SSCR) to improve the gait-based person Re-ID on many multi-covariates. Fendri et al. [46] designed the dynamic selection of human parts (DSHP) to reduce the influence of covariate factors changes. Based on [46], Chtourou et al. [47] futhermore presented a part view transformation model (PVTM) which selects relevant parts through the semantic classification.

**Model-based methods.** Model-based approaches need structured models and human motion models to assist gait recognition. Boucherika et al. [48] extracted features based on the motion model and utilized $K - NN$ classifier as feature clustering. A harr-like template was used for pedestrian detection, while the magnitude and phase of the Fourier components for gait feature extraction. Josinski et al. [49] and Balazia et al. [50] all put forward methods based on the 3D joint information model. The authors of [49] adopted MPCA to reduce the 3D feature dimensionality and the 1-NN algorithm as classifier function, whereas [50] used LDA and PCA for gait features dimensionality reduction and Omaha distance as the similarity metric. When the approaches were tested on the selected feature dataset, the correct classification rate of the two methods is up to 75%.

Although the gait feature is unique, the gait-based person Re-ID methods will hardly obtain a good performance under covariate changes. Appearance features suffer from the covariate changes challenge, while gait features are robust for the covariate variation. On the other hand, appearance features can alleviate the influence of viewpoint changes on gait features. Therefore, appearance and gait feature fusions achieve complementarity, and the fusion feature is more discriminative and robust, which can improve the performance of person Re-ID under covariate changes.

### 2.3. Appearance and Gait-Based Person Re-ID

Appearance features can alleviate viewpoint changes and similarity inter-class problems on gait features and gait features are robust for the covariate changes. Hence, some researchers proposed fusion methods to combine appearance and gait features to achieve complementarity. At present, there are few appearance and gait-based methods of pedestrian recognition. Gala and Shah et al. [12] combined gait feature (GEI/FDEI) with color features to investigate long-term person Re-ID. Liu et al. [13] fused the GEI, color, and texture features at the feature-level and matched the fused features called EIGB with the learning metric. Experiments showed that the proposed method effectively enhanced person Re-ID, but lacked robustness against low-resolution scenarios. Li et al. [10] designed a progressive person Re-ID which consists of appearance-based coarse filtering and gait-based fine search. It employed multi-level features including hand-craft features (CN, HOG, HOG, LOMO) and deep features extracted by GoogleLeNet.

The existing methods regarding appearance features are a combination of multiple manual features or a combination of manual and deep features extracted by GoogleNet, and the multiple feature extraction process is complicated. The gait features lack the optimization to filter out information that is not relevant to a specific pedestrian re-identification task. Meanwhile, the fusion methods based on the two features include score-level fusion, feature-level fusion, and progressive decision fusion. The score-level fusion and

progressive decision fusion in these methods could not avoid the appearance change's effect, and Rank-n matching accuracy is low. In feature-level fusion, they did not consider the information difference, and fused them equivalently. Therefore, the design mode and performance of person Re-ID methods based on feature fusion need to be further improved.

## 3. Research Methodology

### 3.1. Proposed Framework

The proposed framework contains three parts: the first part is the appearance feature extraction branch, the second part is the gait feature extraction branch, and the last is feature fusion. We use the ResNet-50 backbone as an image-level feature extractor for the first branch. For the second branch, human silhouette sequences were extracted from the corresponding image sequence as another input. In the last step, we concatenate the appearance features with standardized gait representation in the feature level. We present the framework for learning representation in a supervised manner and train the model under the guidance of the loss function to achieve better performance.

### 3.2. Appearance Feature Extraction

Typically, video-based person Re-ID methods based on deep learning include image-level feature extractors, temporal modeling modules, and loss functions. Considering that most person Re-ID methods [51–53] adopt the ResNet-50 backbone as a baseline model, we also use the network as an image-level appearance extractor in this paper. The aim of the temporal modeling method for the video-based person Re-ID is to aggregate a sequence of image-level features into a clip-level feature. In this framework, we utilize average pooling to reduce the dimension of features and ensure the complete transmission of information to aggregate image-level features. Compared with the other methods (such as temporal attention, RNN, 3D CNN), the temporal pooling method is simple and efficient [54]. A sequence of the image-level features of $f_c^t, t \in [1, T]$ is a $T \times D$ matrix where $T$ is clip-level sequence length and $D$ is image-level feature dimension. In temporal pooling model, we utilize average pooling to deal with the sequence of image-level features and the result $F_a \in R^D$ formulations are shown as Equation (1),

$$F_a = \frac{1}{T} \sum_{t=1}^{n} f_c^t.$$
(1)

### 3.3. Improved-Sobel-Masking Active Energy Image

We design an ISMAEI algorithm to extract gait features and highlight the contribution of dynamic regions for discriminants and minimize the effect of low-quality silhouettes caused by carrying objects and clothing. We replace the traditional GEI [55] with the AEI method [56] to obtain gait representation. Specifically, when given a gait silhouettes sequences $F = \{f_0, f_i, \ldots, f_{N-1}\}$ in which $f_i$ denotes the $i$th silhouette, and $N$ represents the number of frames in the sequences, we calculate the differences of two adjacent silhouette images to extract active regions, as seen in Equation (2):

$$D_t(x, y) = \begin{cases} f_{t(x,y)} & t = 0 \\ \|f_t(x, y) - f_{t-1}(x, y)\| & t > 0 \end{cases}$$
(2)

$$A(x, y) = \frac{1}{N} \sum_{t=0}^{N-1} D_t(x, y)$$
(3)

where $t$ denotes time and $D_t(x, y)$ is active regions at time $t$ extracted from differences silhouette frames between time $t - 1$ and time $t$. Then, we accumulate these active regions and divide the sum by the numbers of frames $N$ to generate active energy image $A(x, y)$. The final AEI formulation is shown as Equation (3).

Compared with GEI, which keeps both the dynamic and static information, AEI precisely extracts the information of active regions (i.e., the limbs) and throws away information of the static areas (i.e., the trunk of the body). There are still the least dynamic features on the mid-portion of the AEI images. According to intuition, the proper solution is to find the regions of less contribution and mask them out, which can improve the recognition rate. We normalize all AEI and split the body in the AEI into eight parts. The range from the second part under the chin to the fifth part above mid-thigh is defined as the mid-portion. In order to find the least important feature regions and mask them out automatically, we design the Improved-Sobel-masking method, which combines Sobel edge detection operator and soft-threshold wavelet de-noising. Algorithm 1 shows the whole algorithm process of a gait silhouette sequence.

---

**Algorithm 1** Improved-Sobel-Masking Active Energy Image

---

**Require:** A gait silhouettes set $F = \{f_0, \ldots, f_i | i = 0, \ldots, N-1\}$, where $f_i$ denotes $i$th silhouette, $f_i \in R^{m \times n}$; $W$ and $W^{-1}$ denote the two-dimensional orthogonal discrete wavelet transform (DWT) matrix and its inverse, respectively;

**Ensure:** $\widetilde{F}$ denotes a gait representation of the sequence;

 1: **for** $i = 0$ *to* $N - 1$ **do**
 2:    **if** $t = 0$ **then**
 3:       $D_t(x, y) = f_t(x, y)$;
 4:       **Continue**;
 5:    **else**$D_t(x, y) = f_t(x, y) - f_{t-1}(x, y)$;
 6:    **end if**
 7: **end for**
 8: $A(x, y) = \frac{1}{N} \sum\limits_{t=0}^{N-1} D_t(x, y)$, $A \in R^{m \times n}$;
 9: Do DWT $Y = W(A)$, $Y = \{LL_k, HL_k, LH_k, HH_k | k = 1, \ldots J\}$;
10: Estimate the noise variance $\sigma^2 = \left[ \frac{median(|Y_{ij}|)}{0.6745} \right]^2$, $Y_{ij} \in$ subband $HH_1$;
11: Compute the scale parameter $\beta = \sqrt{\log\left(\frac{L_k}{J}\right)}$, $L_k$ is the length of the subband $k$th scale;
12: **for** $i \in \{LH, HL, HH\}$ **do**
13:    **for** $j = 1$ *to* $J$ **do**
14:       Calculate standard deviation $\sigma_y$ of the subband;
15:       Compute threshold $T_N = \frac{\beta \sigma^2}{\sigma_y}$;
16:       Apply soft thresholding $\widehat{X}(i, j) = sgn(x). \max(|x| - T_N, 0)$, $x$ denotes the noisy cofficients of the $i$th class and $j$th scale subband;
17:    **end for**
18: **end for**
19: Invert DWT $\widehat{A} = W^{-1}(\widehat{X})$;
20: $\widetilde{F} = \varphi(\widehat{A})$, $\varphi$ denotes soble edge detection function

---

We tested the algorithm on some samples, and the experimental results are shown in the Figure 3. As can be seen from the figure, the gait presentations that masked the mid-part by Improved-Sobel-Masking are clearer and more discriminative than the others. The Sobel-Masking method performed fine edge detection, but it also leads to excessive noise and unsatisfied feature selection. The Improved-Sobel-Masking operator can eliminate the contour of the knapsack and reduce the influence of external covariates on features. From the above result analysis, we believe the Improved-Sobel-masking method is an excellent choice to post-process the AEI.

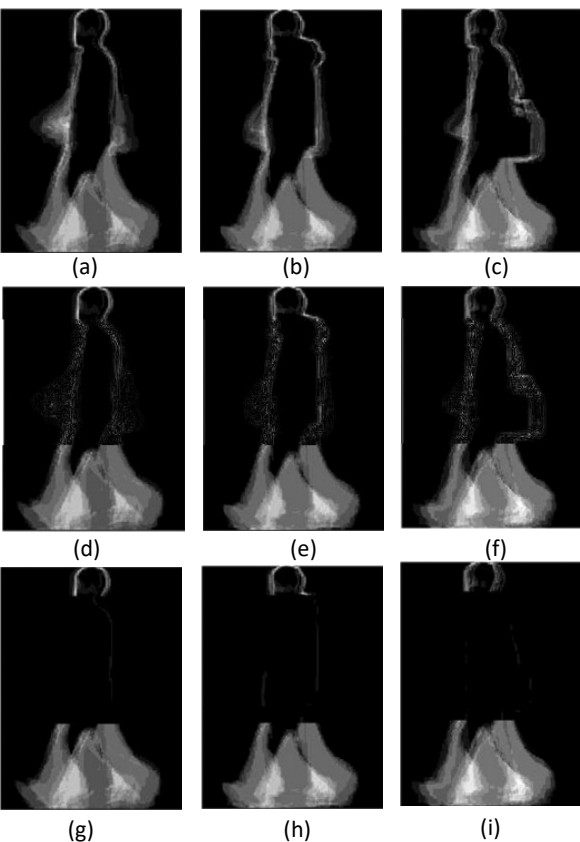

**Figure 3.** AEI and Masking result: (**a–c**) AEI, (**d–f**) Sobel-masking, (**g–i**) Improved-Sobel-masking.

### 3.4. Feature-Level Fusion

There are always three fusion methods: confidence-level fusion [57], feature-level fusion [58], and image-level fusion [59]. In confidence-level fusion, the match scores outputs by multiple biometric matches are consolidated to render a decision about the identity of an individual. Feature-level fusion is that the feature sets originating from multiple biometric sources are integrated into a single feature set by the application of appropriate feature normalization, transformation, and reduction schemes [60]. Image-level fusion combines raw biometric information to account for inter-class and intra-class variability and facilitates decision-making based on the fused raw information. Compared with the other two fusion methods, the primary benefit of feature-level fusion is to combine correlated features extracted from different biometric detection algorithms into a compact set of salient features that can improve recognition accuracy. Therefore, we proposed an improved feature-level fusion method for efficiently combining appearance features with gait features.

There are three classical feature-level fusion methods [61] containing Bitwise add, Hadamard product, and Concatenate. "Bitwise add" represents element-wise pixels of the feature map add which increases the depth of features and does not change the channels. "Hadamard product" is a matrix multiplication operator, which introduces spatial projection and reduces dimension. "Concatenate" means cascading the features, which increases the number of channels of the feature map. These fusion methods can improve the richness and robustness of feature information from different aspects. Therefore, we model and test these three fusion methods to find the best fusion method for the proposed framework. Assuming that we have an appearance feature vector $\vec{F}_a \in R^{M \times N \times Q}$ and a gait feature vector $\vec{F}_g \in R^{M \times N \times Q}$, these fusion methods can be formulated as follows:

$$\vec{F} = f(\vec{F}_a, \theta \vec{F}_g) \tag{4}$$

where the output feature $\overrightarrow{F} \in R^{N \times (2M)}$, and $f$ stands for fusion function. In the formulation, $M$ denotes the number of samples in a batch, $N$ denotes the number of frames, and $Q$ represents the feature dimension. Considering that the gait feature contains trajectory information (the whole gait cycle) and the appearance feature includes the tracklet information, we give the weight $\theta$ to the gait feature and concatenate with the appearance feature, as shown in Figure 4. The final formulation is defined as Equation (5):

$$\vec{F} = \begin{cases} \vec{F}_a \oplus \theta \vec{F}_g \\ \vec{F}_a \odot \theta \vec{F}_g \\ [\vec{F}_a, \theta \vec{F}_g]. \end{cases} \tag{5}$$

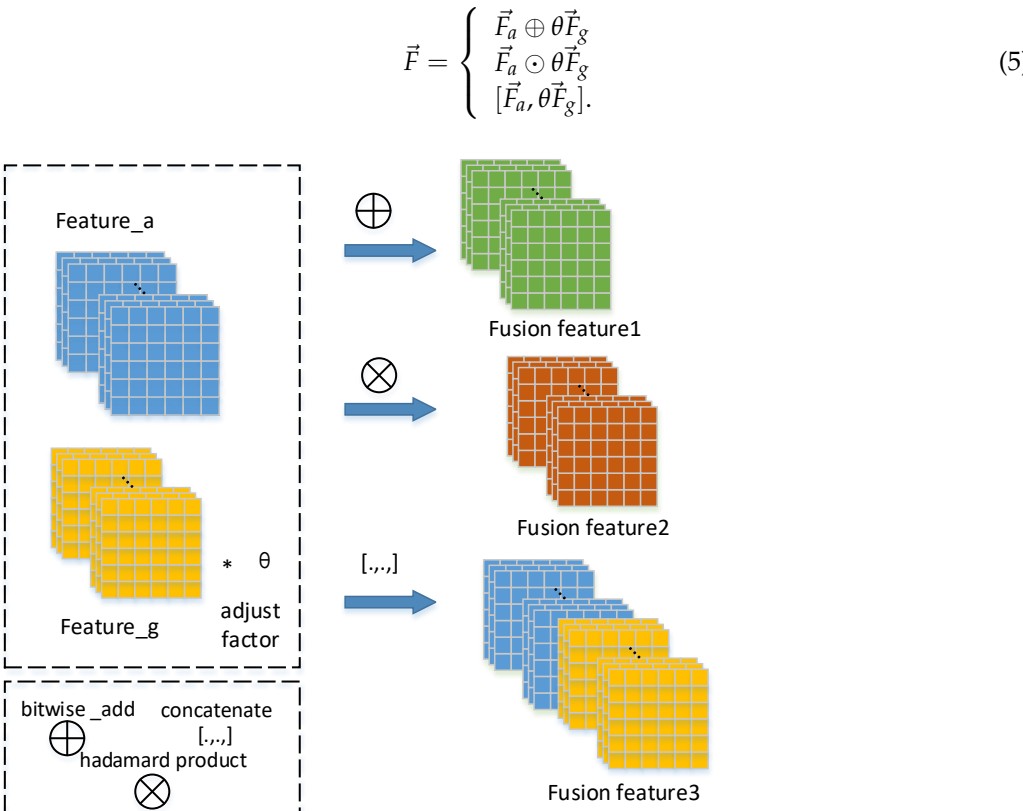

**Figure 4.** The three feature-level fusion modes between appearance and gait features. The sign $*$ denotes multiplication operations.

*3.5. Loss Function*

The proposed framework is trained by batch-hard triplet loss function and cross-entropy label smooth loss function. The batch-hard triplet loss function trains the network to maximize inter-class distance and minimize intra-class distance. In this function, a batch consists of $P \times K$ clips (every clip contains $T$ frames), formed by randomly sampling $P$ identities and randomly sampling $K$ clips for each identity. For a sample $A$ in the batch, the loss formula is defined as follows:

$$L_{tri} = \overbrace{\sum_{i=1}^{P} \sum_{a=1}^{K}}^{\text{all anchors}} \left[ m + \overbrace{\max_{p=1\cdots K} D\left(f_a^i, f_p^i\right)}^{\text{hardest positive}} \right. \\ \left. - \underbrace{\min_{\substack{j=1\cdots P \\ n=1\cdots K \\ j \neq i}} D\left(f_a^i, f_n^j\right)}_{\text{hardest negative}} \right]_+, \tag{6}$$

where the Hinge function $[\cdot]_+ = max(0, \cdot)$, $f_a^i, f_n^i, f_p^i$ denote the $i$-th triplet anchor feature, negative feature, and positive feature. The cross-entropy label smooth loss function $L_{xent}$ utilizes label smooth to reconstruct the $q_i$ of cross-entropy loss, instead of the original

method of $q_i = 1/0$ when $i = y$ or not. The loss function can avoid the Re-ID model from overfitting training and is characterized as Equation (7):

$$
\begin{aligned}
L_{xent} &= \sum_{i=1}^{N} -q_i \log (p_i) \\
q_i &= \begin{cases} 1 - \frac{N-1}{N}\varepsilon & i = y \\ \varepsilon/N & i \neq y \end{cases}
\end{aligned}
\tag{7}
$$

where $y$ is truth ID label and $p_i$ is ID prediction probability of class $i$, small constant $\varepsilon \in [0, 1]$. The parameter $\varepsilon$ can encourage the model to be less confident in the train set. The total loss function $L$ is the combination of the two losses:

$$
L = \alpha * L_{xent} + \beta * L_{tri},
\tag{8}
$$

where $\alpha, \beta$ denotes balanced coefficients of total loss function. We set balanced coefficients $\alpha = \beta = 1$ as final coefficients after experiments. More implementation details can be found in Section 4.2.2.

In summary, the gait silhouettes and traditional GEI gait representation suffer from the covariate changes. Therefore, we design the Improved-Sobel-Masking Active energy image(ISMAEI) algorithm to ensure the consistency of the target pedestrian's gait feature in the situations. We masked dynamic parts of the active energy image by Improved-Sobel masking operator and kept the static regions in the algorithm. Appearance features are less discriminative under covariate change, and the ISMAEI features provide an implicit biometric representation to alleviate the above headache for the Re-ID task. Hence, we fused gait extracted by the ISMAEI algorithm and deep appearance feature extracted by "ResNet50TP" to generate a more robust feature.

## 4. Results and Discussion

### 4.1. Datasets

We tested our proposed method on the CASIA Gait Database B and TUM-GAID dataset, as displayed in the Figure 5a,b. The first dataset contains 124 individuals, which were captured from 11 view angles in three conditions. The view angles contain $0° \sim 180°$ with an interval of $18°$. The three conditions include normal walking (NM), walking with a coat (CL), normal walking with a bag (BG). Each subject has been recorded six times under normal walking, two times under walking with a coat, and two times under normal walking with a bag, so there are six "NM" condition sequences (NM01-NM06), two "BG" condition sequences (BG01, BG02), and two "CL" condition sequences (CL01, CL02) for an individual. The second dataset records 305 individuals' motions from side viewpoints. Each of the 305 persons has ten recorded sequences: Six normal walkings (N1–N6), two backpack variations (B1–B2), and two shoe variations (S1–S2). Furthermore, the 32 people of 305 individuals who underwent ten additional recordings a few months later are called TN1–TN6, TB1–TB2, TS1–TS2. In the experiments, we chose the whole CASIA Gait Database B and the above 32 participants' samples in the TUM-GAID dataset.

We split the datasets into the training and testing sets. In the training set, a complex condition that consists of different states samples from CASIA Gait Database B and TUM-GAID dataset. There are single, cross, and long-term conditions in the testing set. In every condition, there is a probe set that contains the probe images and a gallery set which includes huge images for retrieving. We partition "NM" state samples of ID numbers 85-124 of the first dataset into probe sets and gallery sets in a single condition. In the cross conditions, we divided ID numbers 85-124 of the first dataset into three cases ("CL-NM", "BG-NM", "CL-BG") according to the different states of samples in the probe and gallery set. We divided the remaining 12 participants' samples under the "TB, TN, TS, N" states of the TUM-GAID dataset into probe set and gallery set in the long-term condition. We conducted experiments on these conditions to validate the performance of our method

under the appearance variable and time variable. The detailed experimental settings are shown in Table 1.

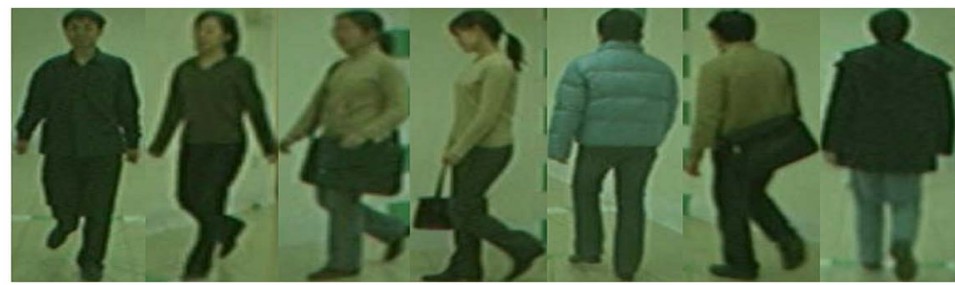

(**a**) CASIA Gait Database B

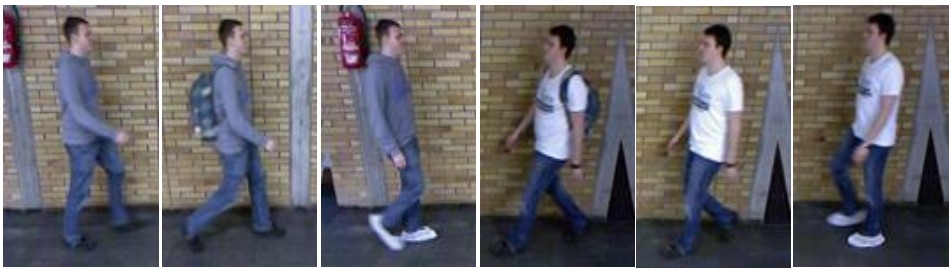

(**b**) TUM-GAID dataset

**Figure 5.** Experimental Datasets (**a**) CASIA Gait Database B include samples under different view angles and states. (**b**) The left three images are captured earlier months than the right. The person has been recorded twice in different conditions in the TUM-GAID dataset.

**Table 1.** Experimental data settings on CASIA Gait Database B and TUM-GAID dataset.

| Sets | Conditions | Covariates | IDs | Sequences (Probe Set \| Gallery Set) | | View Angles (Probe Set \| Gallery Set) | |
|---|---|---|---|---|---|---|---|
| Training set | Complex condition | NM, CL, BG | 1-84 | NM01-NM06, BG01-BG02, CL01-CL02 | | $0$–$180°$ | |
| | | N, B, S,TN, TB, TS | 1-20 | N1-N6, B1-B2, S1-S2, TN1-TN6, TB1-TB2, TS1-TS2 | | $90°$ | |
| Testing set | Single condition | NM-NM | 85-124 | NM01, NM02 | NM05, NM06 | $0°, 90°, 180°$ | $0$–$180°$ |
| | Cross conditions | CL-NM | 85-124 | CL01, CL02 | NM05, NM06 | $0°, 90°, 180°$ | $0$–$180°$ |
| | | BG-NM | 85-124 | BG01, BG02 | NM05, NM06 | $0°, 90°, 180°$ | $0$–$180°$ |
| | | CL-BG | 85-124 | CL01, CL02 | BG01, BG02 | $0°, 90°, 180°$ | $0$–$180°$ |
| | Long-term condition | (TB, TN, TS)-(N) | 21-32 | TN5, TN6, TB1, TB2, TS1,TS2 | N1-N6 | $90°$ | $90°$ |

*4.2. Implementation Details*

4.2.1. Model Parameters and Evaluate Metric

In this paper, the backbone network is standard Res-Net50, and pre-trained model parameters are based on the ImageNet dataset. In the training phase, we chose Adam as optimizer and set parameters: learning rate $lr_k = 0.0001$, learning decay rate $d = 0.1$, weight-decay $\tau = 5 \times 10^{-4}$, and $maxepoches = 400$. The experimental hardware conditions are an Intel i5 CPU and TITAN RTX GPU with 24GB and software conditions are Python v3.6 and Pytorch v1.10.

There are two widely used measurements Cumulative Matching Characteristics (CMC) [62] and mean Average Precision (mAP) [16] in the Re-ID evaluation system. CMC-k (*a.k.a*, Rank-k matching accuracy) [62] means the probability of a correct match appearing in

top-k ranked searching results. Because CMC considers the first match in Re-ID evaluation, the CMC is accurate when there is only one ground truth for each query in the gallery set. Nevertheless, the gallery set includes multiple ground truths in an extensive network, and CMC could not wholly represent the discriminability of a model across multiple cameras. The other widely used indicator is mAP [16] and the metric measures the average retrieval performance with multiple ground truths. In the follow-up experiments, we employed the Rank-n matching accuracy and mAP as the standard measurements.

4.2.2. Performance on Balanced Coefficients of Total Loss Function

To evaluate the impact of balanced coefficients $\alpha, \beta$ in the total loss function Equation (8) for the proposed model, we trained and tested the model according to different balanced coefficient pairs. There are three balanced coefficient pairs $\{(\alpha, \beta)|(1, 1), (0.1, 10), (10, 0.1)\}$, so ratios $\delta$ ($\delta = \frac{\alpha}{\beta}$) between balanced coefficients include $1, 0.01$, and $100$. We considered that the value ranges of the two sub-loss functions may be in different magnitudes and chose the balance coefficient pairs $(\alpha, \beta)$, which better balance the constraints of different loss functions. Meanwhile, we used different ratios $\delta$ to verify which sub-loss function contributes more to the performance of the model.

The training loss curves at different ratios $\delta$ are shown in Figure 6. From the figure, we concluded that when guided by the total loss function which consists of the sub-loss functions by ratios $\delta = 1$, the model converged faster and fitted distributions of the sample better than the models at other ratios $\delta$. At the same time, it proved that the triplet loss function contributed to the model performance more than the smoothing cross-entropy loss function by observing the position and tendency of the blue and red curves. We tested the trained models at three ratios $\delta$ under single and cross conditions and displayed experimental results in Table 2. As can be seen from the table, the model trained at ratios $\delta = 1$ exceeds the models at other ratios in terms of Rank-1 matching accuracy. Therefore, we set $\delta = 1$, i.e., $(\alpha, \beta) = (1, 1)$ as optimal ratio between balanced coefficients for total loss function.

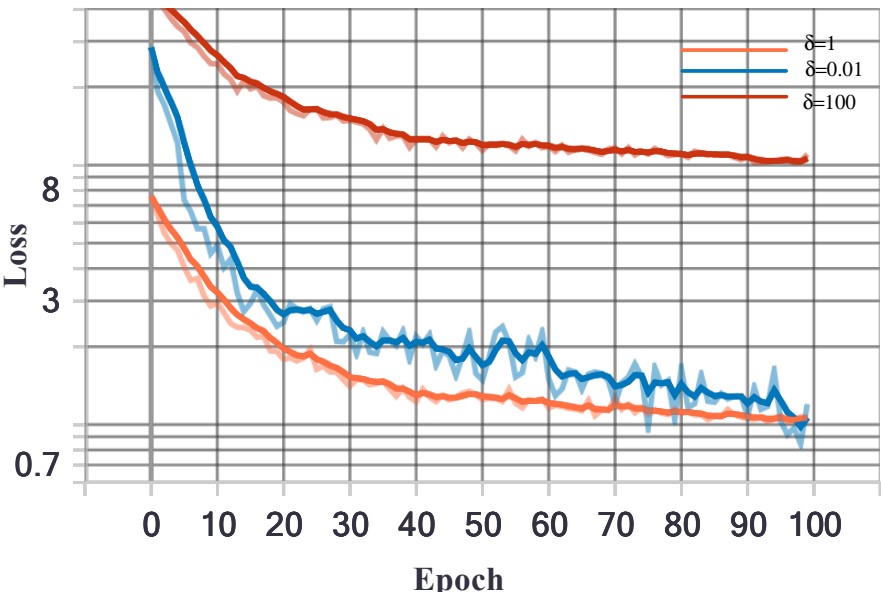

**Figure 6.** Training loss curves on different rations $\delta$ between balanced coefficients $(\alpha, \beta)$ of the total loss function Equation (8). The rations $\delta$ include $1, 0.01$, and $100$ and these ratios denote $(\alpha, \beta)$ is $(1, 1), (0.1, 10), (10, 0.1)$ respectively.

**Table 2.** Rank-1 matching accuracy (%) comparisons among different ratios $\delta$ between balanced coefficients of the total loss function. We trained the proposed model under the supervision of different ratios $\delta$ and tested the trained models on single and cross conditions.

| Rank-1 (%) Conditions | | | | |
|---|---|---|---|---|
| Ratios $\delta$ | NM -NM | CL-NM | BG-NM | CL-BG |
| 1 | **98.2** | **68.5** | **67.5** | **66.9** |
| 0.01 | 87.2 | 66.7 | 60.3 | 60.1 |
| 100 | 74.1 | 60.1 | 57.9 | 56.4 |

### 4.2.3. Feature Visualization

In this part, we used t-SNE [63] to visualize the final learned features for our proposed method and the "GEI + ResNet50TP + L2" method in Figure 7. The "GEI + ResNet50TP + L2" method consists of L2 norm distance and the fused features including GEI gait representations with appearance features extracted by ResNet50. In Figure 7, each color represents an identity that is randomly chosen from the cross conditions in the testing set and the numbers 1 to 10 represent the person's identity numbers. It can be seen that the features of the same identity in Figure 7a stay closely, but most of them are distributed separately in Figure 7b (e.g., person2, person5, person10). From the qualitative analysis perspective, we infer that the GEI representation of the "GEI + ResNet50TP + L2" method still contains covariate changes information, and some features of the same person are far from another. The proposed method adopts SMEAEI gait representations that can overcome the influence of cloth changing and bag or not conditions. We can conclude that the proposed method could better maximize intra-class distance and minimize inter-class distance than the GEI + ResNET50TP + L2 method.

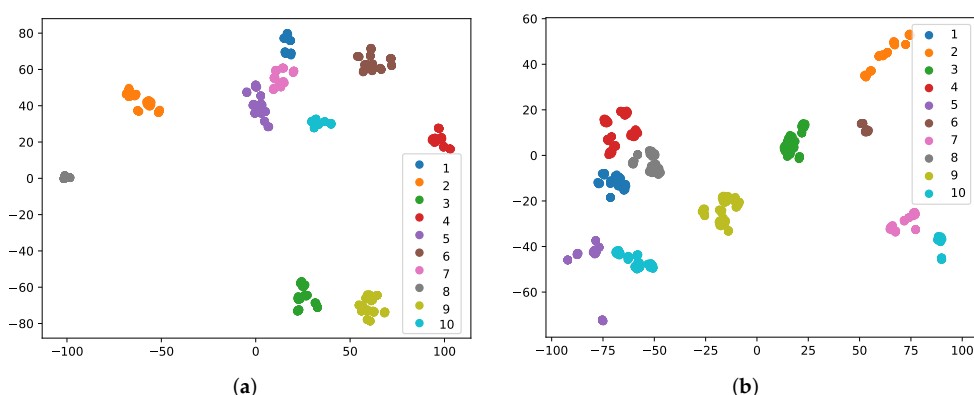

(**a**)   (**b**)

**Figure 7.** The t-SNE visualization for the features (**a**) proposed method features and (**b**) "GEI + ResNet50TP + L2" features on the testing set. Each color represents an identity and the features of every person are extracted from samples under "BG", "CL", and "NM" three condotions.

### 4.2.4. Performance on Adjustment Factor $\theta$

In the fusion part, the appearance and gait features are fused by concatenating. The gait feature includes a whole walking cycle, while the appearance feature contains a video clip. When calculating the similarity among videos, the value of $\theta$ can adjust biometric information to fuse with appearance features and form the discriminate features. We tried to find the optimal value of $\theta$ by enumeration test on the CASIA Gait Database B. We chose the values of the adjustment factor $\theta$ which contain $0.1, 0.3, 0.5, 0.7, 1$ to evaluate the influence of the different value of $\theta$. The experiments were conducted on a single condition and the cross Conditions of the testing set, and the results are shown in Figure 8.

When choosing the optimal value of $\theta$, we consider it from two main aspects: 1. The performance indicator mAP is as large as possible for different theta values; 2. The mAP should be as large as possible under various test conditions. From Figure 8, we can see

that under NM-NM, CL-NM, and CL-BG test conditions, $\theta = 0.5$ satisfies the above two constraints. Although the mAP when $\theta = 0.5$ is slightly less than it when $\theta = 0.1$ by 1.5% under BG-NM conditions, we think the $\theta = 0.5$ is more suitable as the optimal choice from a global perspective.

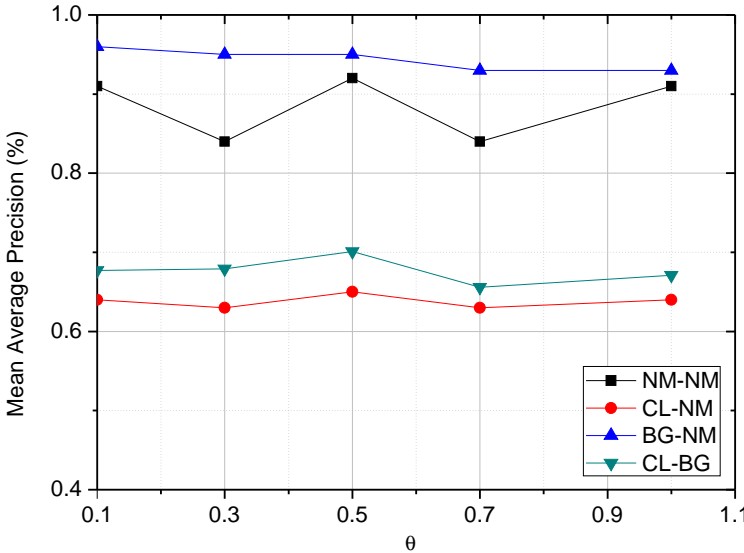

**Figure 8.** The mAP under the single condition and cross conditions with different values of $\theta$.

### 4.2.5. Performance on Different Fusion Methods

Gait features can reduce the impact of covariate changes on appearance features, and appearance features can alleviate the effect of perspective changes on gait representations. We fused the two representations to obtain an informative and discriminative feature. In order to choose the best suitable fusion method for our approach, we tested three fusion methods introduced in Section 3.4 on the cross and long-term conditions. Considering the fusion method designed for overcoming the covariate changes, we did not test the three fusion methods on a single condition where persons keep the same appearance.

The experimental results are shown in Table 3. From the table, the mAP of the proposed approach adopting whatever fusion method represents a clear trend of decreasing when covariate changes gradually become large. However, the "Concatenate" method exceeds the other fusion methods in terms of mAP under cross and long-term conditions. We infer that the cascading method means only concatenating two features together and does not operate on gait feature vectors, so it better ensures the invariance of gait identifies under appearance change and even across periods. Therefore, we choose the cascade method for features fusion in the proposed method.

**Table 3.** Comparison among the three fusion methods on the cross condition and long-term condition of the testing set. mAP (%) is reported.

| mAP (%) Conditions / Methods | CL-NM | BG-NM | CL-BG | (TB, TN, TS)-N |
|---|---|---|---|---|
| Bitwise add | 70.1 | 68.6 | 65.4 | 64.7 |
| Hardmard product | 69.2 | 65.3 | 64.1 | 60.1 |
| Concatenate | **77.7** | **73.3** | **72.5** | **70.9** |

### 4.3. Ablation Study

In this section, we investigate the effectiveness of each component, including the appearance features, gait features, and the fusion in our proposed approach by conducting a series of experiments on the testing set. Specifically, in the first case, we use the "ResNet50TP" network to extract the appearance feature and adopt L2 norm distance as

a metric function, namely "ResNet50TP + L2". The second case, "ISMAEI + L2", adopts ISMAEI gait features and L2 norm distance. The third case, "GEI + ResNet50TP + L2", combines traditional GEI and appearance features. In the last case, The proposed method fuses ISMAEI and appearance features.

As can be seen from the experimental results in Table 4, the performances of the proposed method, the appearance-based method, and the fused other features method are close and have high accuracy under a single condition. We inferred that appearance features based on deep learning keep consistent for the same person when the external covariates remain unchanged. The method based on gait features alone has a low performance because of view angle change. When the external covariates change, the mAP indicators of different methods degrade, but the proposed method outperforms the other methods. Especially, the performance of the proposed method exceeds the "GEI + ResNet50TP + L2" method by 7.2%, in (TB, TN, TS)-N condition where probe and gallery samples have a noticeable difference. The reason for this phenomenon is that our method used ISMAEI to find the invariance of gait representation and suppress the interference information such as clothing changes. Meanwhile, we designed a new fusion method with an adjustment factor $\theta$ to integrate the two features better. Therefore, our method is more robust to the covariate changes, and the experiments proved the effectiveness of our approach.

**Table 4.** Comparison among the different modules of the proposed method and the proposed method on the testing set. mAP (%) is reported.

| mAP (%)  Conditions  Methods | NM-NM | CL-NM | BG-NM | CL-BG | (TB, TN, TS)-N |
|---|---|---|---|---|---|
| ResNet50TP + L2 | 96.0 | 51.1 | 40.8 | 49.7 | 48.1 |
| ISMAEI + L2 | 54.5 | 53.7 | 55.1 | 58.55 | 71.1 |
| GEI+ResNet50TP + L2 | 97.1 | 57.6 | 63.1 | 60.1 | 74.9 |
| proposed method | **98.2** | **63.5** | **67.4** | **66.5** | **82.1** |

### 4.4. Comparison with the State-of-the-Art Methods

In order to better analyze the performance of the proposed method, we further compared the proposed method with state-of-the-art methods including EIDB [13], SSCR [45], HFEI [43], DSHP [46], DG-Net [39], PVTM [47]. Experiments were conducted on the CASIA Gait Database B where the probe and gallery set all contain different view angles and the three states ("NM", "CL", "BG") of the same person.

The comparison results are listed in Table 5. From the qualitative perspective, we can find that the proposed method performs better than those based on gait or appearance features alone. Furthermore, it has more significant outperformance than EIGB (feature-level) combining multiple appearance features and GEI features. From the quantitative perspective, the proposed method achieves 74.2% in terms of Rank-1, which exceeds the second-best methods by 6.9% (from 74.2% to 67.3%) and is lower than HFEI and DG-Net by 0.6% and 0.9% in terms of Rank-5 and Rank-20, which is a small gap. In the Re-ID model performance evaluation, the Rank-1 metric can characterize the matching accuracy and the first hit rate of the model. A high Rank-1 indicator can show a strong characterization ability and fast retrieval. The Rank-$n$ ($n > 1$) is an extension of Rank-1, but it does not reflect the retrieval speed. Therefore, Rank-1 is a relatively critical metric in the Re-ID task. Besides, the mAP measures the average retrieval performance of Re-ID models. Our method outperforms the second-best method by 9.4% (from 72.3% to 62.9%) in terms of mAP. To sum up, the experimental results demonstrate that the proposed method achieves better performance than other methods in the first hit rate and mean retrieval precision.

**Table 5.** Comparison with the state-of-the-art methods on the CASIA Gait Database B. Rank-n matching accuracy (%) and mAP (%) are reported.

| Methods | Rank-*n* Matching Accuracy (%) | | | | mAP (%) |
|---|---|---|---|---|---|
| | **Rank-1** | **Rank-5** | **Rank-10** | **Rank-20** | |
| EIGB (feature-level) [13] | 25.5 | 45.1 | 62.7 | 78.4 | 23.1 |
| SSCR [45] | 67.3 | 87.1 | 91.2 | 93.2 | 60.7 |
| HFEI [43] | 59.1 | 79.1 | 85.7 | **95.1** | 56.3 |
| DSHP [46] | 52.4 | 78.2 | 85.5 | 93.5 | 51.5 |
| DG-Net [39] | 66.9 | **88.1** | 89.9 | 92.7 | 62.9 |
| PVTM [47] | 60.1 | 71.2 | 87.5 | 94.5 | 59.2 |
| Proposed Method | **74.2** | 87.5 | **91.7** | 94.2 | **72.3** |

## 5. Conclusions

For the sake of dealing with the person Re-ID under covariate changes challenge, the paper proposes a novel method that combines appearance features based on "ResNet50TP" and gait features based on the ISMAEI algorithm to person Re-ID task. Firstly, we use ResNet50 to extract the appearance features from video sequences and aggregate the appearance features by the average pooling. Secondly, we adopt the ISMAEI model to process human silhouettes to form gait features. Finally, we combine the appearance and gait features with adjustment factors at the feature-level integration. To test the proposed method's performance, we conducted extensive experiments on the two datasets. Experimental results show that the proposed method exceeds state-of-the-art methods in different data settings.

Although the proposed method performs well, there is room to improve its performance. The ISMAEI is generated by masking AEI with Improved-Sobel-Masking operators, and the gait representation is efficient but less robust to an incomplete gait cycle. So our method's performance degrades in a complicated condition, and we think the skeleton based on deep learning, which could better solve the limitation. Further research is required to make our proposed method closer to the practical application requirements. Hence, we intend to design an end-to-end framework that easily embeds Re-ID solutions into the intelligence video system.

**Author Contributions:** Conceptualization, X.L. (Xiaoyan Lu); methodology, X.L. (Xiaoyan Lu); software, X.L. (Xiaoyan Lu); validation, X.L. (Xiaoyan Lu) and W.S.; formal analysis, X.L. (Xiaoyan Lu); investigation, X.L. (Xiaoyan Lu) and W.S.; resources, X.L. (Xiaoyan Lu); data curation, X.L. (Xiaoyan Lu) and W.S.; writing—original draft preparation, X.L. (Xiaoyan Lu); writing—review and editing, X.L. (Xiaoyan Lu) and W.S.; visualization, X.L. (Xiaoyan Lu); supervision, X.L. (Xinde Li) and S.S.G.; project administration, X.L. (Xinde Li); funding acquisition, X.L. (Xinde Li). All authors have read and agreed to the published version of the manuscript.

**Funding:** This work was supported in part by the project under Grant 2019ZT08Z780, in part by the National Natural Science Foundation of China under Grant 62073072, in part by the Science and Technology on Information System Engineering Laboratory under Grant 05202003, in part by the Key Projects of Key R&D Program of Jiangsu Province under Grant BE2020006 and Grant BE2020006-1, and in part by Shenzhen Natural Science Foundation under Grant JCYJ20210324132202005.

**Conflicts of Interest:** The authors declare no conflict of interest.

## Abbreviations

The following abbreviations are used in this manuscript:

| | |
|---|---|
| ISMAEI | Improved Sobel Masking Active Energy Image |
| Re-ID | Re-Identification |
| GEI | Gait Energy Image |
| AEI | Active Energy Image |
| CNN | convolutional Neural Network |
| LL | Low-Low |
| LH | Low-High |
| HL | High-Low |

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
