# Peer review of "Long-Term Person Re-Identification Based on Appearance and Gait Feature Fusion under Covariate Changes"

_processes, doi:10.3390/pr10040770_

Round 1

Reviewer 1 Report

The article proposed a novel method for long-term person Re-ID based on fusion features. The method was tested on two available databases. The proposal is interesting and it is recognized the contribution to the problems of Re-ID. However, some concerns should be addressed in order to understand the manuscript better and to improve the quality of the paper. 

Methods:

They were tested three fusion methods on the validation set. However, in the Dataset section is unclear how to training and testing data were selected for the three conditions, as included in Table 1. Please, include details about single, Cross, and Long-term conditions.

Table 1 dataset is referred to as Probe set and Gallery set. Please, include a description of these terms. 

The use of training and testing is used in Methods. In the Results section, the term "Validation" is a bit confused. Please, use the same notation for these terms throughout the test. 

Results:

Accuracy, mean average and precision score were used to assess the performance of methods compared. The results should be included standard deviation. 

Also, significant statistics would be desired to understand better the true contribution of the compared methods. In section 4.2.4., the cascade method was chosen only by comparing the accuracy but no information about the distribution of the results is included. The same should be considered for Tables 3 and 4.

The top-rank matching rates presented in Table 4 are not described in methods or results. 

Minor concerns:

- Although AEI was defined in Fig. 2, please, define it also in the text.
- Equation 3. Please, define 'A' in the text.
- Nm and NM are confused throughout the text. Also for CL and Cl; BG and Bg.
- Table 3. Please, indicate in the title the metric presented in the table.

Reviewer 2 Report

Attached the comments

Reviewer 3 Report

This work focuses on an interesting and traditional topic in computer vision -- "person re-identification". But the paper has some flaws. I think some necessary improvements can make this work more convincing.

1. Layout problem.

1) Figure 2 is first mentioned in page 2. But Figure 2 is in page 5.

2) The State-of-the-art Methods mentioned in Section 4 are never discussed in 'Related work'? Are they related to this paper?

3) Section 2 should be improved. See below for details.

2. Weak innovation?

1) Edge detection + denoise = Improved-Sobel-Masking

Both Sobel edge detection and wavelet based denoise are traditional methods. Fig 3 shows some differences. To what degree, can this modification help in the final design? Can the neural network itself makeup this difference? To make the paper more convincing, a more solid analysis may be helpful.

2) More importantly, in Section 2, Appearance-based and Gait-based methods are discussed separately. However, the authors also claim there exist some methods that are

'a fusion of the two features'.

'These methods primarily combine hand-crafted appearance features with gait energy image (GEI) features'.

Analysis about those combined methods are important because the authors claim the proposed technique can optimize the appearance and gait feature and better complement each other'.

If this work is not the first one that proposed the hybrid method, then the innovation is only incremental. If the authors want to highlight their contribution, a further comparison is crucial.

3. Loss function = L_tri + L_xtent.

Why are these two sub-components treated equivalently? Any reasons or considerations?

4. Other minors:

A more careful proof-reading is necessary to enhance the English academic writing. For example,

1) to some certain -- to some degree?

2) have less robust -- are less robust?

3) into two aspects -- from two aspects?

... I can't enumerate them all.

Round 2

Reviewer 1 Report

I thank the authors for their careful response to each one of the concerns in the revision. I believe that each point was considered and solved in the current version and I agree with their comments. Thus, I suggest the publication of the manuscript in Processess.

Author Response

We greatly appreciate your comments to this paper

Thank you very much again!

Reviewer 2 Report

Thanks for the detailed reply, I have a few more questions as follows.
1. A figure is needed to demonstrate that 1 and 1 in the loss function are the best combination of coefficients.
2. In the final comparison of the different methods, the inclusion of other evaluation criteria could also be considered in order to prove the best results of the method proposed in this paper.

Reviewer 3 Report

I still have some questions about your response. It would be great if these can be clarified.

In the response, you mentioned:

"We conducted experiments and validated the effectiveness and superiority of our approach."

"We have conducted several experiments for selecting
cofficients of loss function
"

Can you please clearly point out where your experiment results and analysis can be found in the current paper?
